# Comparative study of artificial light plant factories and greenhouse seedlings of SAOPOLO tomato

Jun Zou[1]*, Wenbin Liu[1], Dawei Wang[1], Shipeng Luo[1], Shaojun Yang[2], Xiaotao Ding[3], Mingming Shi[1]*

1 School of Science, Shanghai Institute of Technology, Shanghai, China, 2 Shanghai Youyou Agricultural Technology Co., Ltd., Shanghai, China, 3 Shanghai Academy of Agricultural Sciences, Shanghai, China

* authorzoujun@sit.edu.cn (JZ); mmshi@sit.edu.cn (MS)

## Abstract

In the summer, the high temperatures, high humidity, frequent rainstorms, and typhoons in the East China region limit the growth of SAOPOLO tomato seedlings. By using a plant factory combined with an LED artificial light environment, the light environment can be effectively controlled to produce high-quality seedlings. This study investigates the growth and energy consumption of tomato seedlings in an artificial light plant factory. The experiment compared tomato seedlings cultivated in the artificial LED light environment of a plant factory with those grown in a semi-enclosed seedling greenhouse. The study meticulously examined the actual growth and development processes of the tomato seedlings, systematically tracking and recording the specific impacts of different cultivation environments on the seedlings' growth and development. Additionally, the experiment followed up on the fruiting conditions of the subsequent tomato plants. The experimental results show that compared to tomato seedlings grown in a greenhouse, those cultivated in the artificial light plant factory grew more slowly before grafting, characterized by slightly lower plant height, relatively smaller leaf area, and slightly thinner stems. However, after grafting, the growth rate of the tomato seedlings in the plant factory significantly accelerated, with increased plant height, leaf area, and stem diameter. On the 16th day after grafting, the cumulative leaf length and width fitting curves for the two cultivation methods coincided. Furthermore, it is noteworthy that the electricity consumption during the tomato seedling cultivation process, including that for controlling environmental temperature and humidity and the LED artificial supplemental lighting in the plant factory, was significantly lower. Over the two-month seedling cultivation period, the resource consumption in the greenhouse was 220% and 281% higher than in the plant factory, respectively. Statistical results also showed that the mortality rate of tomato seedlings cultivated in the artificial light plant factory was only 4.3%, much lower than the 6.5% mortality rate in the greenhouse. When the subsequent tomato plants were uniformly transplanted to the greenhouse for cultivation and their fruit weights were measured and recorded, the results indicated no significant difference in the fruit weights of tomatoes grown in the plant factory compared to those grown in the greenhouse. Therefore, experimental evidence confirms that cultivating

**Data availability statement:** All relevant data are within the article and its Supporting Information files.

**Funding:** This work was supported by the Shanghai Science and Technology Committee (STCSM) Science and Technology Innovation Program, under Grant Nos. 22N21900400 and 23N21900100, awarded to Jun Zou.

**Competing interests:** The authors have declared that no competing interests exist.

tomato seedlings in an artificial light plant factory can significantly reduce cultivation costs, increase seedling survival rates, and not affect tomato quality.

## 1. Introduction

As China's agriculture continues to develop, the demand for vegetable seedlings is steadily increasing. According to the Food and Agriculture Organization (FAO), China's tomato production reached 70 million tons in 2023, accounting for more than one-third of the world's total output. However, in the East China region, the process of tomato seedling cultivation is often hindered by the summer environment characterized by high temperatures, high humidity, frequent rainstorms, and typhoons, which result in low temperatures and weak light. These conditions limit seedling growth, leading to reduced yield and quality. The artificial light environment in plant factories can create suitable conditions for seedling growth, minimizing or eliminating the impact of adverse natural environmental factors. Seedlings within the system experience shorter growth cycles and less pollution [1]. By combining year-round multi-shelf cultivation, the land utilization rate and seedling yield per unit area can be increased by several times or even dozens of times [2]. Plant factories equipped with artificial lighting create a controlled environment, which can increase crop yield and enhance the efficiency of land, water, energy, and nutrient use [3]. They have always shown great potential for stable and efficient production of horticultural product [4,5]. Throughout the entire plant growth cycle, light controls the physiological responses and growth development of plants [6,7]. Precise control of the light environment for crop cultivation can effectively improve seedling survival and crop yield [8,9]. In this experiment, based on the optimal light-quality ratio of tomato seedlings determined in previous studies, a large-scale nursery was carried out on this basis to explore its feasibility.

As a new variety introduced from France, SAOPOLO tomato has significant advantages in varietal characteristics, although its cultivation conditions are relatively demanding [10,11]. (添加引用)This variety has strong disease resistance and wide adaptability, which can reduce the use of pesticides, lower production costs, and improve economic benefits. At the same time, it aligns with the modern agricultural development trends of being green, ecological, and safe. Secondly, the promotion and application of SAOPOLO tomatoes contribute to advancing agricultural technology innovation and the modernization process. The cultivation of this variety employs advanced soilless cultivation techniques and comprehensive digital management. Combined with the artificial light environment plant factory, it represents the development direction of modern agricultural technology [12]. The promotion and application of SAOPOLO tomatoes can facilitate the dissemination and application of agricultural technology, driving the upgrade and transformation of the agricultural industry. Additionally, with plant lengths exceeding ten meters and nearly two hundred lateral branches per plant, the yield of SAOPOLO tomatoes is significantly higher than that of ordinary tomatoes. Traditional cultivation methods rely heavily on chemical pest control and may lack standardization and precision in cultivation techniques and management, which can negatively impact tomato growth and yield [13]. By utilizing plant factories for seedling cultivation, these issues are perfectly resolved. Intelligent management and precise environmental control significantly increase seedling survival rates while reducing production costs.

Additionally, LEDs are the first light sources with controllable spectra. By adjusting their spectrum, the wavelength of LED light can be matched to the plant photoreceptors, thereby enhancing plant yield and quality [14]. This study employs intelligent LED supplemental lighting systems with adjustable light quality, intensity, and photoperiod, which can maximize

light energy utilization efficiency. Ricardo et al. found that under red and blue LED light, the biomass accumulation, leaf area, and chlorophyll content of tomatoes were all greater than under monochromatic red or blue LEDs. Specifically, at an R:Bof 2.3, the growth efficiency of tomatoes was 172% higher compared to that under cool white fluorescent lamps [15]. This study used SAOPOLO tomato seedlings cultivated in different environments within an artificial light plant factory and conducted experiments in the following steps: (1) Measured the growth data of seedlings before grafting to compare the environmental impacts. (2) Measured the data of tomato seedlings during the seedling stage to compare the environmental impacts. (3) Recorded the survival rate of tomato seedlings and energy consumption during the seedling period in both environments. (4) Counted the number and quality of mature fruits in both environments.

## 2 . System design and design of experiments

### 2.1. Design of artificial light seedling plant factory system

In this study, we constructed a hybrid combination of vertical farming and an intelligent artificial light plant factory. The artificial light plant factory is equipped with an intelligent environmental control system, while the vertical plant supplemental lighting framework consists of a four-layer setup, integrated with an intelligent light environment control system.

**2.1.1. Structure of the seedling plant factory.** The artificial light environment experimental plant constructed in this study consists of an environmental monitoring and intelligent control system as well as a vertical plant supplemental lighting frame equipped with an intelligent light environment control system, which monitors the environmental parameters, temperature, humidity, carbon dioxide concentration, and light environment parameters of the supplemental lighting frame through sensors, and then real-time regulation and control through the main control panel to ensure that the plants are stabilized in an optimal growth environment. (as shown in Fig 1). The environmental monitoring and control system utilizes a wireless sensor network and field server-based monitoring to enable real-time monitoring and regulation of relevant environmental parameters through application software. The vertical supplemental lighting framework consists of four layers, each equipped with 12 LED light strips. Each lighting framework is integrated with its own control system. The light uniformity on each layer exceeds 70%, and the Photosynthetic Photon Flux Density (PPFD) can reach over 200 μmol/m²·s (as shown in Fig 2).

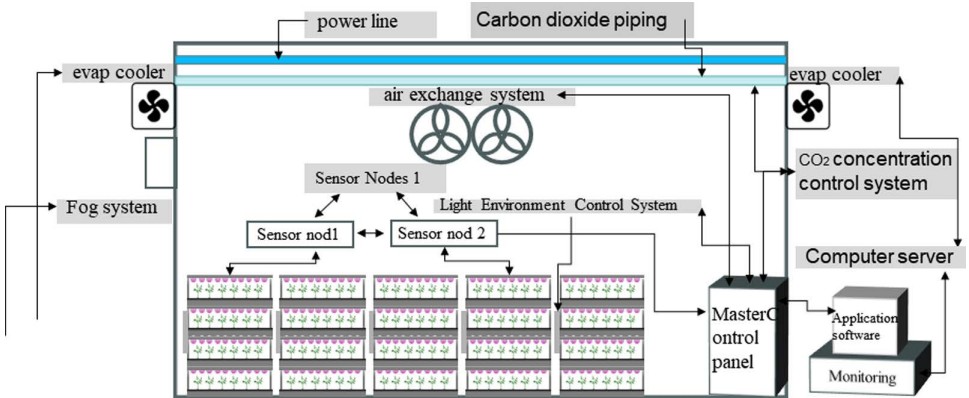

**Fig 1.** Artificial light environment plant factory and its components.

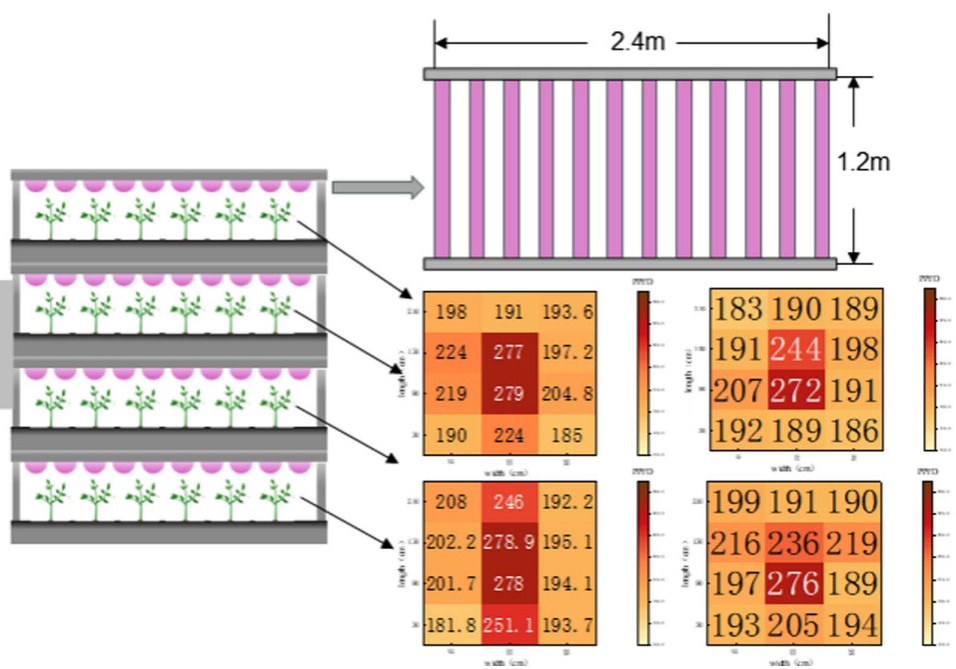

**Fig 2.** Structure of the vertical supplemental lighting framework.

**2.1.2. Seedling plant factory control system.** Based on the research results of the light regulation mechanism of tomatoes and other plants, a data fusion control system was constructed to achieve precise regulation of the crop light environment (as shown in Fig 3). This system includes three main components: the light environment monitoring subsystem, the data analysis subsystem, and the decision-making subsystem. The light environment monitoring subsystem mainly monitors the light quality, PPFD and photoperiod in real time and records the real-time data, which is then passed to the data analysis subsystem to get the target value of light environment regulation at the current planting time. According to the real-time light quality and PPFD, the difference between the light quality and the dimming value is calculated, and the dimming signal is sent to the LED supplemental light device for driving or dimming. By integrating these subsystems, the data fusion control system ensures precise and dynamic adjustment of the light environment, optimizing the conditions for tomato plant growth and enhancing overall crop quality and yield.

The system controller is the core of the intelligent LED plant supplemental lighting system. Based on the actual needs of the plant factory, the design of the main control part follows a modular approach. The main components include the main controller, slave controllers, storage module, power meter module, and operation interface. The block diagram of this system part is shown in Fig 4.

According to the above system design and hardware construction, the physical setup of the overall design of the artificial light plant factory is shown in Fig 5.

## 2.2. Materials and methods

**2.2.1. Plant material and growing conditions.** The experiment selected the commercial hybrid tomato variety 'Saopolo' as the scion and 'Maxifort' as the rootstock. Seeds of 'Saopolo' and 'Maxifort' were directly sown into rock wool plugs (cylindrical,

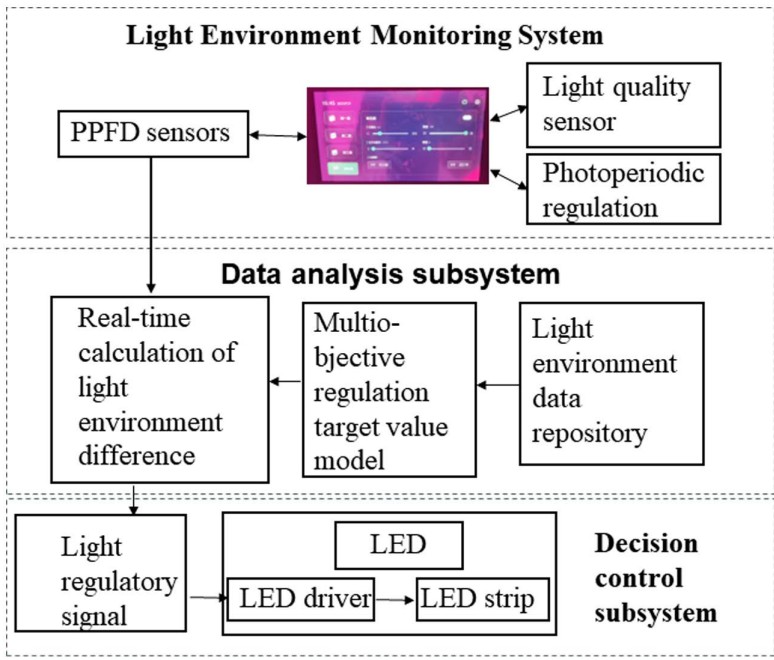

**Fig 3.** Block diagram of light environment closed-loop regulation system.

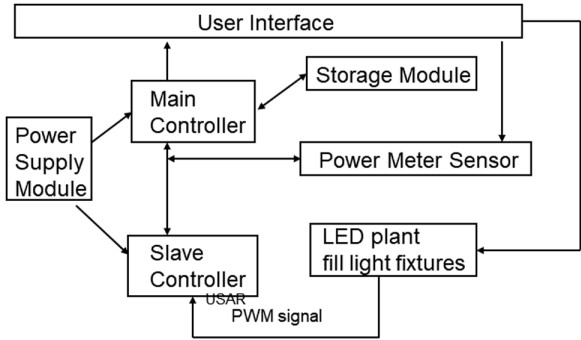

**Fig 4.** Block diagram of the system controller section.

20 mm in diameter, and 27 mm in height), then covered with a vermiculite substrate, and germinated in a germination room at 25°C. The rock wool plugs in the trays (240 holes) were thoroughly soaked with water, with an EC value of 1.5–2.0 dS/m, and a pH value of 5.5–6.0. The relative humidity of the germination chamber was maintained at 85% or more. The trays were then placed in a seedling greenhouse (where high-pressure misting and air conditioning maintained daytime temperatures at 25–30°C and air conditioning maintained nighttime temperatures at 23–25°C) until 50% of the plants were visible (approximately 2–3 days). After 15 days of sowing, when both the scion and rootstock had developed two true leaves, half (120 plants) of the rootstock seedlings were transplanted to another tray, ensuring an even distribution across two trays. After the tomato seeds germinated, they were watered with a nutrient solution prepared using a Japanese horticultural formula. The primary element concentrations in mol/L were N: P: K: Ca: Mg = 16:4:8:8:4, and the trace

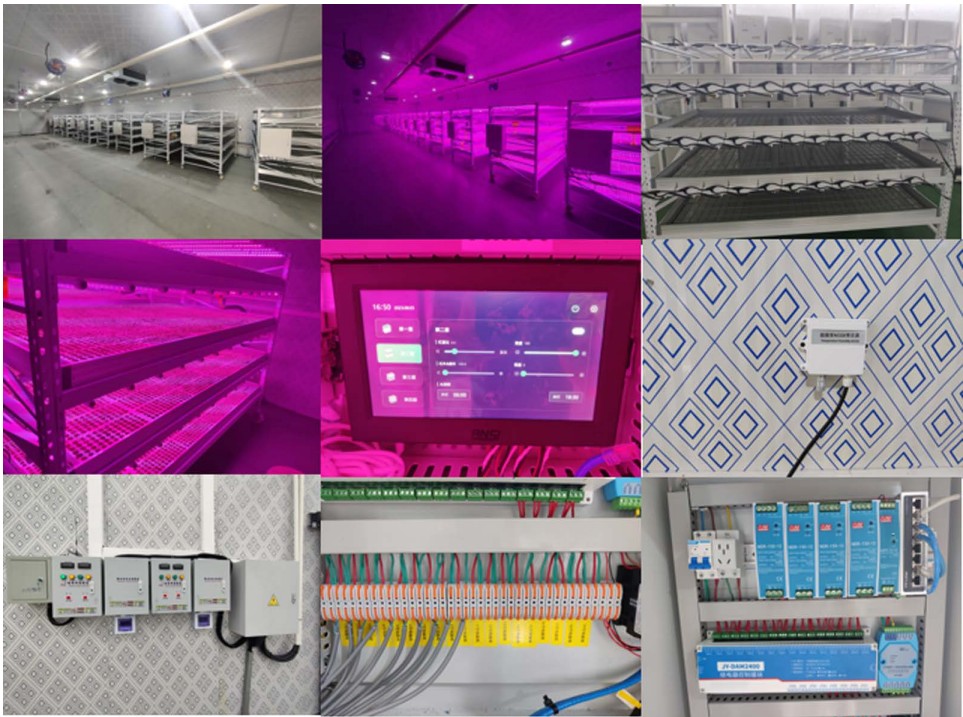

**Fig 5.** Physical drawing of part of artificial light plant factory.

element concentrations in mol/L were Fe: B: Mn: Zn: Cu: Mo = 3:0.5:0.5:0.05:0.02:0.01. From seed germination to the one true leaf stage, a half-strength nutrient solution was applied every 2 days using bottom irrigation. After the one true leaf stage, a standard concentration nutrient solution with a pH of 6.8 to 7.2 and an EC of 2.0–2.4 mS/cm was used for irrigation. During bottom irrigation, the substrate absorption time for the nutrient solution was more than 30 minutes.

 **2.2.2. Experimental environment design.** In the experiment on the effect of artificial light environments on tomato seedling growth, an artificial light environment using red and blue LEDs with an R：B of 8:2 was applied. During the experiment, a fiber optic spectrometer was used to measure the spectral distribution from 300 nm to 800 nm at a distance of 15 cm below the lights. In all light treatments, the light intensity was maintained at approximately 250 μmol/m²·s.

 **2.2.3. Plant growth data measurement.** *Measurement of plant morphology and growth characteristics*: Starting on the eighth day after tomato seed sowing, 15 tomato seedlings were randomly selected from different positions, divided into three groups of five plants each. Using a steel ruler, the plant height from the surface of the substrate to the tip of the meristem, as well as the length and width of the leaves, were measured in centimeters. Each measurement was taken three times, and the average value was recorded. The stem diameter at the hypocotyl and the middle of the cotyledons was measured using an electronic caliper, in millimeters, with three measurements averaged. On the 16th day after tomato grafting, the number of leaves and the length of the lateral branches were measured, again taking the average of three measurements. During the fruit-setting stage, five tomato plants from each of the two seedling methods were randomly selected. Every week, the diameter of the first fruit cluster on three different lateral branches was measured using a steel ruler, with three

measurements averaged. At the harvest stage, the measured fruits were weighed, with each fruit weighed three times and the average weight recorded.

*Seedling survival and energy consumption for cultivation*: On the fourth day after tomato seedling grafting, the survival rates of 1200 seedlings cultivated under natural light and artificial light environments were recorded. This was done by randomly selecting ten trays of tomato seedlings from the total of 80,000 seedlings and calculating the survival rate for each environment. To measure energy consumption, the cumulative electricity usage for the entire seedling period was recorded by reading the independent electricity meters for the greenhouse and the plant factory. The data collected on energy consumption and survival rates were then analyzed and visualized using Origin 7.5 software (Origin Lab, Northampton, MA).

**2.2.4. Statistical model development.** To estimate the developmental process of different vegetables, we established models of cumulative leaf length (the sum of all leaf lengths per plant) and cumulative leaf width (the sum of all leaf widths per plant) based on the number of days after grafting. All data were analyzed and regressed using SAS statistical analysis software version 9.3 (SAS Institute Inc., Cary, NC), and the results were plotted using Origin 7.5 software (Origin Lab, Northampton, MA).

# 3. Experimental results

## 3.1. Plant factory environmental data

As shown in Fig 6, the environmental monitoring system equipped with the artificial light plant factory recorded environmental data such as ambient temperature, humidity, and carbon dioxide concentration during the tomato seedling nursery process. The data showed that the ambient temperature was maintained at 24-26°C, the humidity in the nursery was maintained at over 85%, and the ambient carbon dioxide concentration was maintained at 450-500 ppm. the stable and suitable nursery environment ensured the quality of tomato seedlings and the accuracy of the experimental results.

## 3.2. Measurement of seedling growth parameters

As shown in Figs 7 and 8, before the grafting of tomato seedlings, the true leaf length of seedlings cultivated in the greenhouse was greater than that of seedlings grown in the plant factory, while the stem diameter was the opposite in the early stages. The reason for this phenomenon is that during summer seedling cultivation, the greenhouse provides more abundant

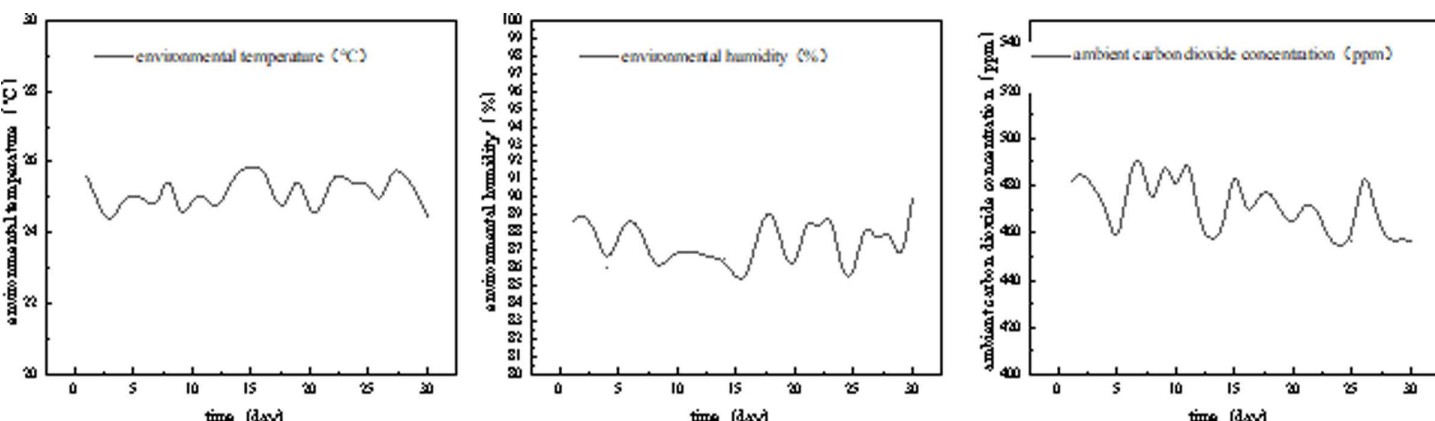

**Fig 6.** 30-Day environmental monitoring data for plant factories.

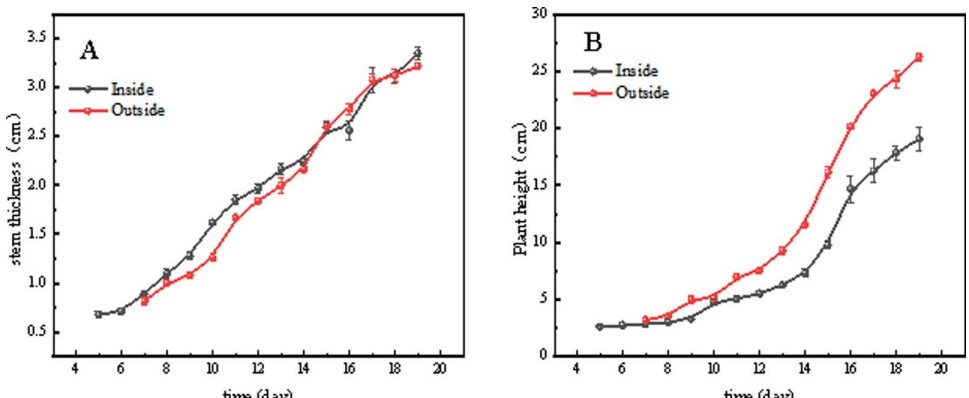

**Fig 7.** (A–C) Leaf length and (D–F) leaf width of the first to third true leaves of the two nursery methods in plant factories and greenhouses.

**Fig 8.** ( A) Stem thickness and (B) plant height of tomato seedlings grown by two nursery methods, plant factory and greenhouse.

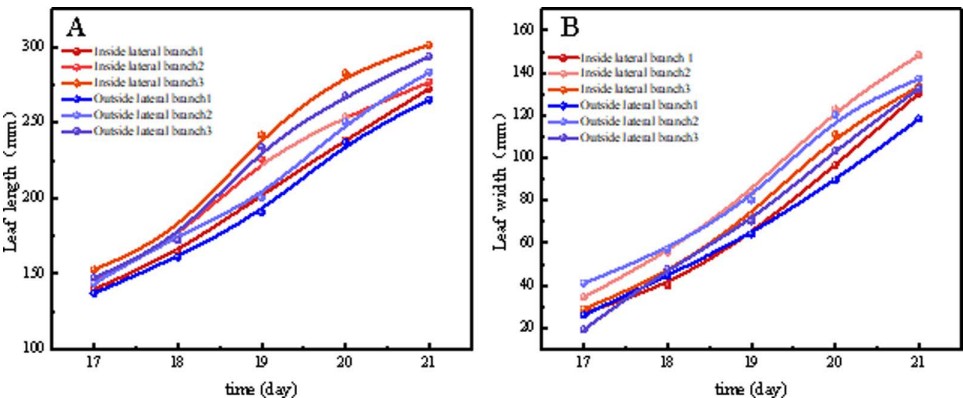

**Fig 9.** Comparison of different (A) lateral branch lengths and (B) lateral branch widths of tomato in the two nursery methods.

sunlight and higher temperatures on sunny days, which favor the expression of certain growth characteristics of SAOPOLO tomatoes, such as the elongation of true leaves. In contrast, the plant factory offers constant environmental conditions that are conducive to stable seedling growth but may not promote the expression of certain growth characteristics.

### 3.3. Growth at true leaf stage

As shown in Fig 9, tomato seedlings grown in the plant factory had better growth rates of all three lateral branches at the true leaf stage than those grown in the greenhouse. In general, red light promoted tomato lateral branch length and stem thickness, while blue light had the opposite effect [16]. This is consistent with a previous study that showed that a higher ratio of blue to red light decreased tomato lateral branch length and stem thickness [17]. However, referring to the previous monochromatic light experiment, the stem thickness of Cuty tomatoes grown under blue light was 1.9 times that of those grown under red light [18], based on this result, a high blue light ratio should promote the growth of tomato lateral branches [19], however, in contrast, the increased blue light ratio resulted in a decrease in lateral branch length. This suggests that tomato plants may be sensitive to a specific ratio of red and blue light rather than to the amount of red and blue light in the combined red and blue LED treatment. This is consistent with the fact that in this experiment, the plant factory selected an R8B2 ratio of LED light quality, which resulted in better growth of tomato lateral branches than tomato seedlings cultivated in the greenhouse (Fig 9A,B). When tomato seedlings were exposed to LED irradiation with a red to blue ratio of 8:2 versus natural light, stem thickness was more affected [20], which is consistent with the results of this study (Fig 10). In addition, Nanya et al. observed greater stem length in tomato seedlings treated with low blue ratio (1B:9R) compared to high blue ratio (5B:5R) treatment [21]. The photoreceptors of cryptophyte pigments are known to be blue light (390-480 nm) receptors that inhibit hypocotyl elongation when stimulated [22,23]. Thus, the inhibition of stem length by blue led is thought to be due to stimulation by cryptogamic pigments (Fig 9A).

### 3.4. Cumulative leaf length and width fitting curve

As shown in the Fig 11 below the correlation between the seedling environment and the cumulative leaf length and width is represented by the correlation coefficient $R^2$. In the regression analysis of cumulative leaf length and width against grafting days, the fitted curves all have values $R^2$ greater than 0.99, indicating a good fit. From Fig 10, it can be seen that there is

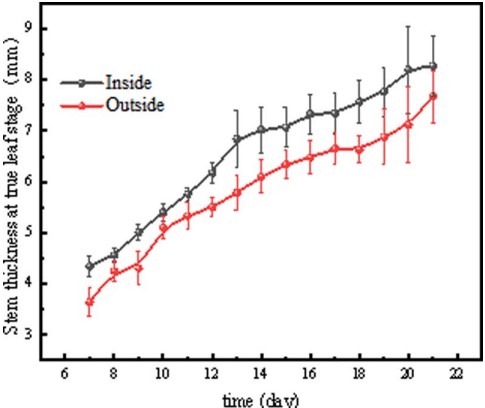

**Fig 10.** Comparison of stem thickness at true leaf stage between the two nursery methods.

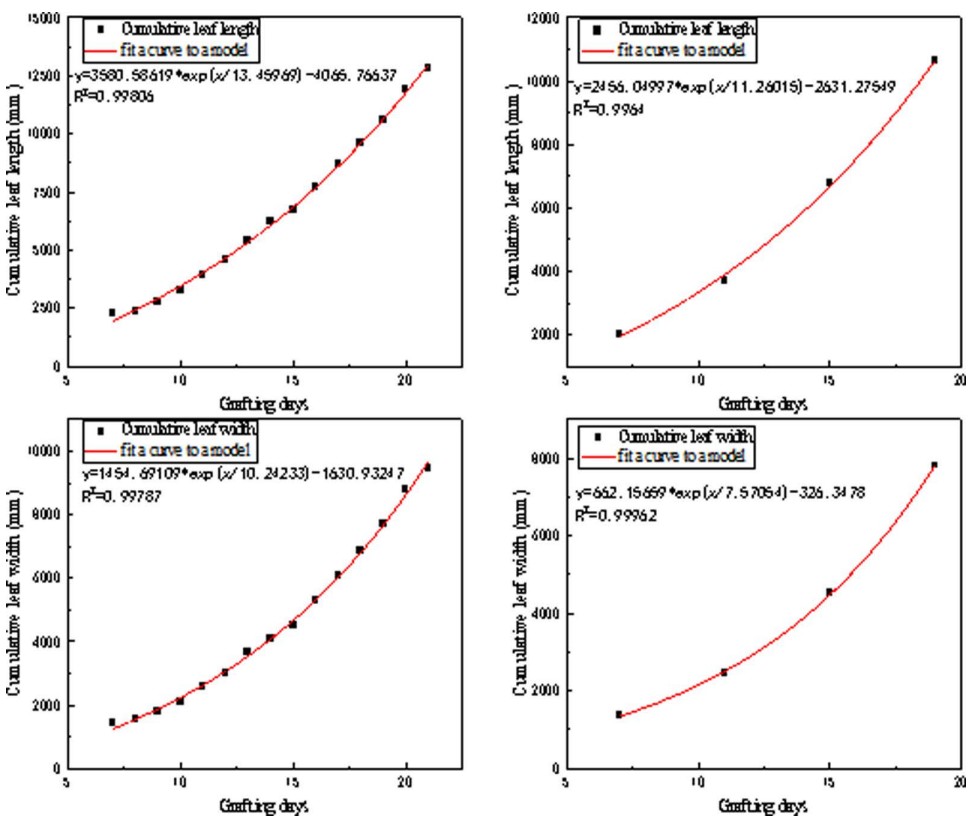

**Fig 11.** Regression analysis of cumulative leaf length and width with days of grafting.

no significant difference in the cumulative leaf length and width of tomatoes under two different cultivation methods. In this study, cumulative leaf length and width are predicted based on developmental days. Some studies consider incorporating light radiation into leaf area accumulation models, such as thermal efficiency and photosynthetically active radiation [24], as well as the light-temperature function [25]. In this study, the greenhouse light is environmental light, and growth depends on external radiation. Light does not always have a positive

effect on leaf area accumulation [26]. Excessive light may have an inhibitory effect on plant leaf growth, resulting in a decrease in leaf area and an increase in leaf thickness [27]. This is consistent with the experimental results of this study. However, supplementary light can increase leaf area and biomass accumulation in winter [28]. Voutsinos et al. pointed out that during the vegetative growth phase, the relative growth rate increases with temperature, while during the linear growth phase, the maximum growth rate is more constrained by radiation than by temperature. Additionally, plant factories have strong temperature control capabilities, providing more effective references for year-round production management based on the growth and development of tomato seedlings using growing days as a basis [29].

## 3.5. Seedling survival and nursery costs

A suitable light environment for seedling growth is the key to increase its survival rate. In photosynthesis, chloroplasts mainly absorb red and blue light [30]. By analyzing the spectra of four commonly used light sources, it was found that grafted tomato seedlings developed well under R8:B2 due to the abundance of red and blue light in R8:B2. Earlier studies have found that red light plays a role in the accumulation of chlorophyll, carotenoids, and anthocyanins [31] and delays flower differentiation and restores internode elongation [32]. In addition, red light helps plants to resist abiotic and biotic stresses [33]. In addition, red light helps to increase plant biomass, while blue light inhibits internode elongation and lateral growth and prevents overgrowth of the plant. In other studies, root formation in isolated seedlings of red palm was induced gradually under red LED light [34]. Solano et al [33] reported that pea and watermelon seedlings exposed to red light for 15 min showed the greatest increase in fresh weight and fresh height, and longer exposure time reduced seedling growth. Exposure of grafted tomato seedlings to different light qualities was statistically different in stem length, with red light exposure being the highest.Kim and Hwang confirmed that plant plant 'Mini Chal' tomatoes under a mixture of blue and red light resulted in high quality [23,32,35]. In addition, the barrier tissue cells in the leaves are particularly well developed and the sponge tissue cells show an organized morphology under red + blue [23,31,36]. Studies on tomato and sage. The results showed that mixed red and blue light can increase the net photosynthetic rate, as well as dry weight and leaf area [36–38]. This coincides with the results of this experiment, as shown in the statistics of Figs 12 and 13, the survival rate of tomato seedlings cultivated in the plant factory during the seedling stage reached 96.3%, which is a significant increase of 3% in the survival rate compared with those cultivated in the greenhouse

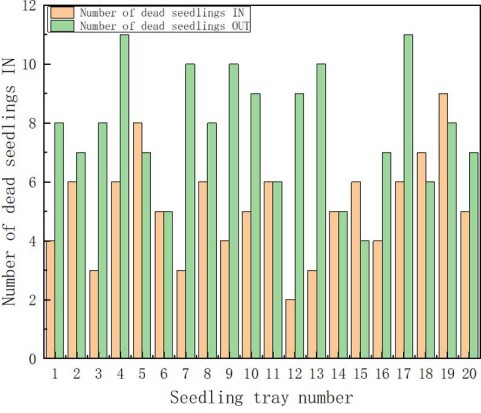

**Fig 12.** Statistics on the number of tomato seedling deaths in the two nursery methods.

greenhouse. At the same time, the electricity consumption of tomato seedlings for two months showed that during the two months of seedling cultivation, the resource consumption of greenhouse was 220% and 281% higher than that of the plant factory.

### 3.6.  Measurement of fruit parameters in ripening tomatoes

As shown in Fig 14, there was no significant difference in the quality of the fruits produced on the three lateral branches of the seedlings raised in the two ways from the time of fruit set to ripening across the different days, but the tomato plants seeded by the plant factory (right) showed a more uniform increase in fruit mass throughout the growth process.

## 4.  Discussion

### 4.1.  Impact of PPFD and light quality

Experiments have shown that PPFD significantly impacts the early seedling growth of tomatoes [39,40]. For cherry tomato seedlings (Solanum lycopersicum cv. Mill qian - xi), an increase in PPFD from 50 μmol m$^{-2}$ s$^{-1}$ to 500 μmol m$^{-2}$ s$^{-1}$ resulted in reduced stem length [41]. The results of the study showed that Momotaro Fight tomato grew and developed more

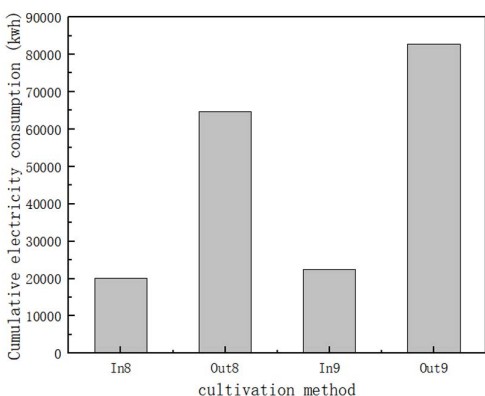

**Fig 13.**  Schematic diagram of electricity consumption during seedling rearing for the two seedling methods.

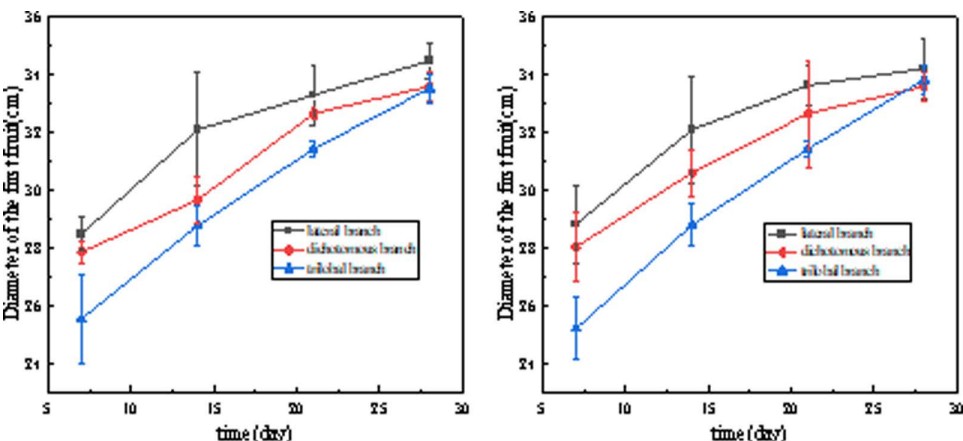

**Fig 14.**  Schematic diagram of fruit diameter for both nursery methods.

rapidly with greater plant height growth under 150 μmolm$^{-2}$s$^{-1}$PPFD light environment compared to 300 μmol m$^{-2}$ s$^{-1}$PPFD light environment [42]. He et al. discovered that for the 'Mill qian - xi' variety, plant height increased as PPFD rose from 50 μmol m$^{-2}$ s$^{-1}$ to 300 μmol m$^{-2}$ s$^{-1}$, but decreased when PPFD further increased to 550 μmol m$^{-2}$ s$^{-1}$ [43]. Different varieties and their stem lengths respond variably to PPFD under different experimental conditions. In this experiment, the natural high PPFD of summer in East China positively influenced the stem elongation of SAOPOLO tomatoes. Typically, the photosynthetic rate of leaves is lower under high PPFD conditions and higher under low PPFD conditions [44,45]. Prolonged exposure to excessive light can produce large amounts of reactive oxygen species, overwhelming the antioxidant system's capacity and causing irreversible photo oxidative damage to chloroplasts and cells, thereby inhibiting photosynthesis [46,47]. This explains why tomato seedlings cultivated in greenhouses have a lower seedling robustness index and higher mortality rates. Light quality can alter plant morphology through photoreceptors and signal transduction systems [48,49]. Compared to blue light, red light is known to increase the length of the hypocotyl and plant height [43,50,51]. In the experiment, plants grown under R4B1 light, which has a higher red light proportion, showed significantly higher dry mass compared to those grown under natural light. This result is visually observable through the comparison of stem thickness between the two types of tomato seedlings. This finding aligns with observations that a high percentage of blue light can inhibit dry matter production [52,53].

### 4.2. Factors affecting tomato fruit quality

There was no significant difference in tomato yield obtained by the two cultivation methods, which may be due to the cultivation method rather than the quality of the lighting, because less water can be used in the peat substrate, which may reduce the weight of the fruits but increases the concentration of active substances and improves the saturation of flavor [54]. Tomato yield and quality are affected not only by the intensity of supplemental light, but also by the quality of the supplemental light. The data suggest that an increase in the amount of red light contributes to the increase in fresh weight of tomatoes, but does not affect the increase in dry matter content. It appeared that red light stimulated an increase in water content in tomatoes. In contrast, the increase in blue light decreased the dry matter content of all tomato varieties [55].

## 5. Conclusion

In this study, we investigated the effects of an artificial light plant factory on the cultivation of SAOPOLO tomatoes. The results showed that during the seedling stage, cultivating SAOPOLO tomatoes in the plant factory significantly improved the stability of seedling growth. This led to more uniform and robust seedlings, reducing issues such as excessive stem elongation, low chlorophyll content, and weak seedlings. The mortality rate of SAOPOLO seedlings during the seedling stage was significantly reduced in the plant factory. This is likely due to the stable environment provided by the plant factory, which ensures continuous and optimal growth conditions for the tomato seedlings. Furthermore, electricity consumption statistics over the two-month seedling cultivation period indicated that cultivating tomato seedlings in the plant factory greatly reduced production costs and energy consumption during the seedling stage. Therefore, seedling rearing in an artificial light environment plant factory can significantly improve the survival rate of tomato seedlings while reducing the cultivation energy consumption under the premise of ensuring the quality of tomato fruits.This study provides valuable reference and insights for large-scale modern artificial light plant factory tomato seedling cultivation.

## Supporting information

**S1 Table. Data measured in graphs and charts in articles**
(DOC)

## Author contributions

**Conceptualization:** Jun Zou.

**Data curation:** Jun Zou, Wenbin Liu, Mingming Shi.

**Formal analysis:** Jun Zou, Wenbin Liu, Mingming Shi.

**Funding acquisition:** Jun Zou.

**Investigation:** Dawei Wang, Xiaotao Ding.

**Methodology:** Wenbin Liu, Dawei Wang, Shipeng Luo, Xiaotao Ding.

**Project administration:** Wenbin Liu, Shipeng Luo, Xiaotao Ding, Shaojun Yang.

**Resources:** Wenbin Liu, Dawei Wang, Shipeng Luo, Mingming Shi, Shaojun Yang.

**Software:** Jun Zou, Wenbin Liu, Dawei Wang, Shipeng Luo.

**Supervision:** Jun Zou, Wenbin Liu, Mingming Shi, Xiaotao Ding.

**Validation:** Jun Zou, Xiaotao Ding.

**Visualization:** Jun Zou, Shipeng Luo, Mingming Shi.

**Writing – original draft:** Wenbin Liu.

**Writing – review & editing:** Wenbin Liu, Mingming Shi.

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
