## [Decision Letter · Decision Letter 0]

15 Oct 2024

Dear Dr. Zou,

Thank you for submitting your manuscript to PLOS ONE. After careful consideration, we feel that it has merit but does not fully meet PLOS ONE’s publication criteria as it currently stands. Therefore, we invite you to submit a revised version of the manuscript that addresses the points raised during the review process.

We look forward to receiving your revised manuscript.

Kind regards,

Yuan Huang

Academic Editor

PLOS ONE

Journal Requirements:

2. Thank you for your submission to PLOS ONE. We note that your cover letter mentions " submission to  Journal of Optics and PhotonicsResearch", but this was submitted to PLOS journal as " submitted to PLOS ONE ". Can you please clarify this and upload the correct file.

3. We note that your Data Availability Statement is currently as follows: [All relevant data are within the manuscript and its Supporting Information files.

Reviewers' comments:

Reviewer's Responses to Questions

**Comments to the Author**

1. Is the manuscript technically sound, and do the data support the conclusions?

Reviewer #1: Yes

2. Has the statistical analysis been performed appropriately and rigorously?

Reviewer #1: Yes

3. Have the authors made all data underlying the findings in their manuscript fully available?

Reviewer #1: Yes

4. Is the manuscript presented in an intelligible fashion and written in standard English?

Reviewer #1: Yes

Reviewer #1: The MS entitle; Comparative study of artificial light plant factories and greenhouse seedlings of SAOPOLO tomato is very productive and i recommended minor revision.

Please add reference in following;

As China's agriculture continues to develop, the demand for vegetable seedlings is steadily increasing. According to the Food and Agriculture Organization (FAO), China's tomato production reached 70 million tons in 2023, accounting for more than one-third of the world's total output. However, in the East China region, the process of tomato seedling cultivation is often hindered by the summer environment characterized by high temperatures, high humidity, frequent rainstorms, and typhoons, which result in low temperatures and weak light. These conditions limit seedling growth, leading to reduced yield and quality. The artificial light environment in plant factories can create suitable conditions for seedling growth, minimizing or eliminating the impact of adverse natural environmental factors.

Please revise this reference

By combining year-round multi-shelf cultivation, the land utilization rate and seedling yield per unit area can be increased by several times or even dozens of times[1];

Please check this reference or might br reference have some mistake

3. Ramin Shamshiri, R.; Kalantari, F.; C. Ting, K.; R. Thorp, K.; A. Hameed, I.; Weltzien, C.; Ahmad, D.; Mojgan Shad, Z.; 1. Smart Farming Technology Research Center, Department of Biological and Agricultural Engineering, Faculty of Engineering,Universiti Putra Malaysia, 43400, Serdang, Selangor, Malaysia; 2. Department of Landscape Architecture, Faculty of Design and Architecture, Universiti Putra Malaysia, 43400, Serdang, Selangor, Malaysia; et al. Advances in Greenhouse Automation and Controlled Environment Agriculture: A Transition to Plant Factories and Urban Agriculture. Int J Agr Biol Eng 2018, 11, 1–22, doi:10.25165/j.ijabe.20181101.3210.

This sentience need to be modify; Converts light energy into carbohydrates[5]. Growers can increase crop productivity[6,7].

Add reference; As a new variety introduced from France, SAOPOLO tomato has significant advantages in varietal characteristics, although its cultivation conditions are relatively demanding.

Please description to Fig. 1 to make easy to understand.

From Figure 10, it can be seen that there is no..... Replace with......as presented in Fig. 10, there is no.......

Add reference............>>>Light does not always have a positive effect on leaf area accumulation.

>irradiation with a red to blue ratio of 8:2 versus natural light, stem thickness was more affected[16], which is consistent with the results of this study (Figure 10). In addition, Nanya et al. observed greater stem length in tomato seedlings treated with low blue ratio (1B:9R) compared to high blue ratio (5B:5R) treatment[17]. The photoreceptors of cryptophyte pigments are known to be blue light (390-480 nm) receptors that inhibit hypocotyl elongation when stimulated[18,19]. Thus, the inhibition of stem length by blue led is thought to be due to stimulation by cryptogamic pigments (Fig. 9A).

As mentioned in above statements, that red to blue ratio of 8:2 versus natural light, stem thickness was more affected,

Can you explain, how and way it effective?

Strong light can even reduce leaf area and increase leaf thickness[22]. This is consistent with the experimental results of this study.

Please revise this sentence.

Voutsinos et al. pointed out that during the vegetative growth phase, the relative growth rate increases with temperature, while during the linear growth phase, the maximum growth rate is more constrained by radiation than by temperature.

Revise this reference and statement.

Revise whole MS and make sure all reference are correct.

Add reference

Experiments have shown that PPFD significantly impacts the early seedling growth of tomatoes.

Modify the following statement

Similarly, Matsuda et al. found that the stem length of the tomato variety 'Momotaro Fight' was significantly longer under 150 µmol m-2 s −1 PPFD compared to 300 µmol m−2 s −1 PPFD[35].

Conclusion

Therefore, seedling cultivation in the plant factory can enhance seedling survival rates and reduce cultivation energy consumption while ensuring fruit quality.

Please revise as; ……..>>>> reduce cultivation energy consumption, and ultimatly improving tomato production and fruit quality.

**Do you want your identity to be public for this peer review?** For information about this choice, including consent withdrawal, please see our Privacy Policy

Reviewer #1: No

---

## [Author Response · Author response to Decision Letter 0]

13 Nov 2024

Accompanying letter

Dear Editor and Reviewer,

We sincerely thank the editors and all the reviewers once again for their valuable feedback. We will use this feedback to further improve our manuscript “Comparative study of artificial light plant factories and greenhouse seedlings of SAOPOLO tomato”. All reviewers' comments have been laid out below in italicized font.Our responses are provided in regular font, while modifications/additions to the manuscript are indicated in red font.

1.Question

Please add reference in following;

As China's agriculture continues to develop, the demand for vegetable seedlings is steadily increasing. According to the Food and Agriculture Organization (FAO), China's tomato production reached 70 million tons in 2023, accounting for more than one-third of the world's total output. However, in the East China region, the process of tomato seedling cultivation is often hindered by the summer environment characterized by high temperatures, high humidity, frequent rainstorms, and typhoons, which result in low temperatures and weak light. These conditions limit seedling growth, leading to reduced yield and quality. The artificial light environment in plant factories can create suitable conditions for seedling growth, minimizing or eliminating the impact of adverse natural environmental factors.

Response:

Thank you for the reviewer's suggestions.In the sections you have pointed out, we have added relevant literature to support our exposition and to provide readers with more comprehensive background information.Once again, we would like to thank the editors and reviewers for their careful review and valuable suggestions on our article. We believe that the quality of the article has been further improved by these revisions. We will be more than happy to discuss and improve it if there are any further issues.The revised section can be found on lines 42-51, highlighted in red font.

2.Question

Please revise this reference:

By combining year-round multi-shelf cultivation, the land utilization rate and seedling yield per unit area can be increased by several times or even dozens of times[1]

Response:

We have re-screened more appropriate references in the relevant sections to ensure the accuracy and rigor of the content. The replaced references better support the arguments of the article.The revised section can be found on lines 51-52, highlighted in red font.

3.Question

Please check this reference or might br reference have some mistake:

3. Ramin Shamshiri, R.; Kalantari, F.; C. Ting, K.; R. Thorp, K.; A. Hameed, I.; Weltzien, C.; Ahmad, D.; Mojgan Shad, Z.; 1. Smart Farming Technology Research Center, Department of Biological and Agricultural Engineering, Faculty of Engineering,Universiti Putra Malaysia, 43400, Serdang, Selangor, Malaysia; 2. Department of Landscape Architecture, Faculty of Design and Architecture, Universiti Putra Malaysia, 43400, Serdang, Selangor, Malaysia; et al. Advances in Greenhouse Automation and Controlled Environment Agriculture: A Transition to Plant Factories and Urban Agriculture. Int J Agr Biol Eng 2018, 11, 1–22, doi:10.25165/j.ijabe.20181101.3210.

Response:

We have re-screened more appropriate references in the relevant sections to ensure the accuracy and rigor of the content. The replaced references better support the arguments of the article.The revised section can be found on lines 54-55, highlighted in red font.

4.Question

This sentience need to be modify; Converts light energy into carbohydrates[5]. Growers can increase crop productivity[6,7].

Response:

Thank you for reviewing our paper and for your valuable comments. In response to the language presentation problems you pointed out, we have carefully and embellished to ensure the accuracy and judgment of the content of the paper.The revised section can be found on lines 57-58, highlighted in red font.

5.Question

Add reference; As a new variety introduced from France, SAOPOLO tomato has significant advantages in varietal characteristics, although its cultivation conditions are relatively demanding.

Response:

In the sections you have pointed out, we have added relevant literature to support our exposition and to provide readers with more comprehensive background information.The revised section can be found on lines 61-62, highlighted in red font.

6. Question

Please description to Fig. 1 to make easy to understand.

Response:

Thank you for your review and feedback on our paper. In response to your question about the interpretation of the images, we have provided additional clarification based on your suggestions.The revised section can be found on lines 101-107, highlighted in red font.

7. Question

From Figure 10, it can be seen that there is no..... Replace with......as presented in Fig. 10, there is no......

Response:

We have modified the specific statements you mentioned, resulting in a more logical correction. We hope that these adjustments will enhance the reader's understanding of the content of the paper.

8. Question

Add reference............>>>Light does not always have a positive effect on leaf area accumulation.

Response:

In the sections you have pointed out, we have added relevant literature to support our exposition and to provide readers with more comprehensive background information.

9. Question

>irradiation with a red to blue ratio of 8:2 versus natural light, stem thickness was more affected[16], which is consistent with the results of this study (Figure 10). In addition, Nanya et al. observed greater stem length in tomato seedlings treated with low blue ratio (1B:9R) compared to high blue ratio (5B:5R) treatment[17]. The photoreceptors of cryptophyte pigments are known to be blue light (390-480 nm) receptors that inhibit hypocotyl elongation when stimulated[18,19]. Thus, the inhibition of stem length by blue led is thought to be due to stimulation by cryptogamic pigments (Fig. 9A).

As mentioned in above statements, that red to blue ratio of 8:2 versus natural light, stem thickness was more affected,

Can you explain, how and way it effective?

Response:

In this experiment, tomato seedlings cultivated in greenhouse greenhouses, because of high summer temperatures and long sunshine hours in East China, led to the phenomenon of seedling growth in the growth process, resulting in slender stalks; while seedlings cultivated in artificial light plant factories can effectively inhibit the phenomenon of seedling growth through the precise regulation of the light environment as well as the temperature and humidity, which significantly improves the growth of the seedlings and increases the rate of strong seedlings.At the same time, red light is more efficient for photosynthesis, which provides more energy for plant growth. This energy accumulation translates into cell expansion, especially stem growth. Thus, red light not only promotes the vertical elongation of cells, but also the horizontal growth of stems, resulting in an increase in stem diameter.

10. Question

Strong light can even reduce leaf area and increase leaf thickness[22]. This is consistent with the experimental results of this study.Please revise this sentence.

Response:

Thank you for reviewing our paper and for your valuable comments. In response to the language problems you pointed out, we have made serious revisions and added references to ensure the accuracy and correctness of the paper's content.The revised section can be found on lines 260-262, highlighted in red font.

11. Question

Voutsinos et al. pointed out that during the vegetative growth phase, the relative growth rate increases with temperature, while during the linear growth phase, the maximum growth rate is more constrained by radiation than by temperature.

Revise this reference and statement.

Response:

In response to the problems you pointed out with the references and statements, we have made careful revisions to ensure the accuracy and scientificity of the content. In terms of references, we have replaced some inappropriate literature and selected more scientific and reasonable references; in terms of statements, we have carefully examined the expressions in the text and optimized the relevant statements in the text in order to improve the accuracy and fluency of the text.The revised section can be found on lines 260-262, highlighted in red font.

12. Question

Add reference

Experiments have shown that PPFD significantly impacts the early seedling growth of tomatoes.

Response:

In the sections you have pointed out, we have added relevant literature to support the ideas presented in the text and to increase the scientific validity and accuracy of the article.The revised section can be found on lines 311-312, highlighted in red font.

13. Question

Modify the following statement

Similarly, Matsuda et al. found that the stem length of the tomato variety 'Momotaro Fight' was significantly longer under 150 µmol m-2 s −1 PPFD compared to 300 µmol m−2 s −1 PPFD[35].

Response:

Thank you very much for your help and guidance, we have revised the specific statements you mentioned, resulting in more logical as well as more accurate language embellishments, and hope that these adjustments will enhance the reader's understanding of the content of the paper.The revised section can be found on lines 314-316, highlighted in red font.

14. Question

Conclusion

Therefore, seedling cultivation in the plant factory can enhance seedling survival rates and reduce cultivation energy consumption while ensuring fruit quality.

Please revise as; ……..>>>> reduce cultivation energy consumption, and ultimatly improving tomato production and fruit quality.

Response:

Thank you for reviewing our paper and for your valuable comments. In response to the specific problems you pointed out with the statements, we have revised and embellished them to increase their accuracy and scientific validity.The revised section can be found on lines 355-357, highlighted in red font.

15. Question

Thank you for your submission to PLOS ONE. We note that your cover letter mentions " submission to  Journal of Optics and PhotonicsResearch", but this was submitted to PLOS journal as " submitted to PLOS ONE ". Can you please clarify this and upload the correct file.

Response:

I am very sorry for adding trouble to your journal due to the error of software operation, I can confirm that the manuscript has been submitted only in your journal, and the correct cover letter has been added in the attachment at the back, so please check it out!

Additionally, in the attached Excel spreadsheet, we have provided all the raw data needed to include the results of the replication study. This includes the values behind the reported means, standard deviations, and other metrics, as well as the values used to construct the graphs.

---

## [Editor Report · Decision Letter 1]

18 Nov 2024

Comparative study of artificial light plant factories and greenhouse seedlings of SAOPOLO tomato

PONE-D-24-42369R1

Dear Dr. Zou,

We’re pleased to inform you that your manuscript has been judged scientifically suitable for publication and will be formally accepted for publication once it meets all outstanding technical requirements.

Kind regards,

Yuan Huang

Academic Editor

PLOS ONE

Additional Editor Comments (optional):

accept
---

## [Editor Report · Acceptance letter]

PONE-D-24-42369R1

PLOS ONE

Dear Dr. Zou,

I'm pleased to inform you that your manuscript has been deemed suitable for publication in PLOS ONE. Congratulations! Your manuscript is now being handed over to our production team.

Kind regards,

on behalf of

Dr. Yuan Huang

Academic Editor

PLOS ONE